# Price elasticity of demand for cigarettes in Nepal: Evidence from a lower middle-income country in South Asia using Deaton's demand model

Vishnu Prasad Sapkota[1]*, Umesh Prasad Bhusal[2], Nirmal Kumar Raut[3]

**1** Department of Economics, Tribhuvan University, Nepal Commerce Campus, Min Bhavan Marga, Kathmandu, Nepal, **2** Health Economics Researcher, Kathmandu Nepal, **3** Central Department of Economics; Tribhuvan University, Central Campus, Kirtipur, Kathmandu, Nepal

* vishnu.sapkota@ncc.tu.edu.np, visapkota@gmail.com

## Abstract

### Background

In South Asia, taxation policies are popular instruments to reduce high cigarette demand which work through directly affecting cigarette prices. The effectiveness of such policies depends on consumers' behavioural responses to changes in cigarette price, which is commonly expressed in terms of the price elasticity of demand. A limited number of studies in this region have used available cross-sectional and panel survey data to estimate cigarette price elasticity using Deaton's demand model, and to date, no such study exists for Nepal. Therefore, this study aims to estimate the price elasticity of cigarette demand in Nepal.

### Methods

We applied Deaton's demand model to estimate cigarette price elasticity utilizing data from the Nepal Household Risk and Vulnerability Survey, a three-year panel micro-data (2016–2018). The analysis is based on data from 5,653 households observed over three waves. Deaton's demand model exploits spatial variation in unit values of cigarettes to measure price elasticity. We adjusted the basic model for possible bias that may arise from panel nature of the data.

### Results

In this study, we found the estimate of price elasticity for cigarettes to be −0.58 [95% CI: −0.79, −0.37]. This indicates that the demand for cigarettes is relatively inelastic. Similarly, we found the estimate of expenditure elasticity to be 1.06 [95% CI: 0.54, 1.57], indicating a more responsive cigarette demand with respect to income.

**Data availability statement:** The data were collected by the World Bank. The authors do not have permission to share the data used for the analysis. The data can be obtained on 02/07/2022 after registration at https://micro-data.worldbank.org/index.php/catalog/3705.

**Funding:** The author(s) received no specific funding for this work.

**Competing interests:** The authors have declared that no competing interests exist.

## Conclusion

The negative price elasticity estimate indicates that the increase in excise taxes can be effective both in reducing cigarette consumption and raising tax revenues. This also provides additional evidence that routine surveys can be valuable for estimating price elasticity using Deaton's method, which are originally designed to monitor living standards.

---

## 1. Introduction

Cigarette smoking is declining globally; however, its prevalence remains highest in South Asia. Recent evidence reports that 27.1% of people aged 15 years and above in South Asia smoke cigarettes—exceeding the global average of 22.3%—with Nepal reporting an even higher prevalence of 30.4% [1,2]. A high smoking prevalence is a major modifiable risk factor for leading non-communicable diseases [3,4]. Beyond its impact on health, smoking worsens living standards and poverty [5–7]. These health and economic impacts, together with the transnational nature of the tobacco industry in South Asia, pose multifaceted challenges for tobacco control [8]. The Framework Convention on Tobacco Control (FCTC) recognises demand reduction of tobacco consumption as an important theme and suggests interventions such as taxation, health warnings, and advertising bans to progress in this agenda [9].

Taxation is a widely employed policy instrument for reducing tobacco consumption, including cigarette smoking [10]. The policy directly raises cigarette prices, which has the potential to achieve the dual goals of improving public health by reducing cigarette demand and raising government revenue [11,12]. However, the policy effectiveness depends on how consumers respond to changes in cigarette prices. A common measure of such behavioural responses is the *price elasticity of demand,* which shows how consumers adjust their purchasing behaviour to price changes, reflecting their preferences, perceived value or necessity of the product.

In the literature, the price elasticity of demand for cigarettes is typically estimated using either time-series or cross-sectional data. The latter is frequently adopted in low- and middle-income countries (LMICs) through the application of Deaton's methodology [13]. This approach provides a rigorous framework for estimating the price elasticity of regular commodities when long-term time-series data are unavailable and/or incomplete. It uses rich data available from large-scale cross-sectional surveys that are primarily designed to monitor living standards and consumer prices.

The price elasticity of cigarette demand in Nepal presents a useful case study for the policy makers in South Asia for several reasons. First, this study represents a pioneer study in Nepal to estimate the price elasticity of cigarette demand using Deaton's method [13]. Also, existing regional evidence is limited and provides mixed conclusions: in India, initial estimates indicate a lower price elasticity of demand for cigarettes (−0.179) compared to *bidis* and leaf tobacco (−0.87) particularly in urban areas [14]. Although subsequent studies showed an increase in the elasticity—ranging from −0.26 for the wealthiest group to −0.83 for the poorest group [15,16]—most

of this evidence is restricted to specific subgroups such as youth. While studies in Bangladesh and Pakistan are based on recent survey data [17,18], evidence from other South Asian countries remains scant despite the availability of rich microdata on tobacco consumption in this region [8]. Second, the prevalence of cigarette smoking in South Asia exceeds the global average, characterized by a diverse range of products and a transnational industry presence [8]. Third, cigarette prices in purchasing-power-parity (PPP) dollar are broadly comparable across the region—standing at 7.85 in Nepal, 8.66 in India, 11.96 in the Maldives, and 2.91 in Bangladesh [19]. Furthermore, while tax rates are similar, no country in the region met the WHO-recommended threshold of a 75% tax share of the retail price in 2022 [20]. Fourth, the cigarette industry in Nepal contributes significantly to public revenue and receives government support, driven in part by concerns regarding employment and investment [8,21]. Fifth, although non-cigarette tobacco products account for more than 60% of total tobacco consumption, they contribute less than 20% of the total tobacco tax revenue [8,22].

Table 1 shows the trend in cigarette pricing and taxation in Nepal over a ten-year period. From 2012 to 2022, cigarette prices increased at a constant rate of approximately 1.00 USD (in PPP) annually. This trend indicates that households encountered a reasonably constant price increment both leading up to and during the survey period. Throughout this same period, excise taxes remained largely stable, fluctuating between 26% and 30% of the retail prices. These statistics suggest that both price adjustments and tobacco taxation remained relatively steady during the study period.

In this context, this study aims to estimate the price elasticity of cigarette demand in Nepal by applying Deaton's demand model [13]. This approach provides a theoretical and procedural framework for estimating price elasticity from cross-sectional surveys, which are frequently utilized to monitor poverty and living standards in LMICs. In these surveys, households report both expenditure and physical quantities of cigarettes—both of which are affected by the prevailing market prices. These two reported metrics are used to derive unit values by dividing *expenditure* by *quantity*. These unit values cannot be treated as proxies for prices, as they are inherently biased by *measurement error* and *quality effects*. Because these surveys observe clusters of households simultaneously, there is often no genuine price variation within a cluster. On the other hand, significant spatial variation typically exists between clusters. The Deaton method exploits this inter-cluster variation in prices to strip away the *measurement error* and *quality effects* from *unit values,* thereby yielding a consistent estimate of the price elasticity of cigarettes. The following section elaborates on this estimation and correction methodology.

## 2. Methodology

In this section, we first elaborate on Deaton's methodology for measuring the price elasticity of demand for conventional commodities. We then apply this approach to estimate the demand equation for cigarette consumption using panel survey data from Nepal. Furthermore, we outline the supplementary assumptions and the adjustments made to the original model—initially designed for cross-sectional surveys—to accommodate the panel structure of our data. Finally, we provide a detailed overview of the data sources, clusters composition, and variable definitions.

**Table 1. Cigarette price and tax in Nepal between 2012 and 2022.**

| Year | 2012 | 2014 | 2016 | 2018 | 2020 | 2022 |
|---|---|---|---|---|---|---|
| Retail price (per 20 sticks in PPP dollars) | 3.65 | 4.69 | 5.63 | 6.78 | 8.38 | 10.43 |
| Retail price (per 20 sticks in equivalent NPR*) | 95 | 130 | 176 | 211 | 266 | 342 |
| Taxes as percentage of cigarette price (%) | 29.86 | 27.79 | 26 | 30 | 27 | 31.39 |

**Source**: The data was accessed from https://www.who.int/data/gho/data/themes/topics/raise-taxes-on-tobacco [23].

* The PPP conversion factors for NPR were obtained from https://data.un.org/Data.aspx?d=WDI&f=Indicator_Code%3APA.NUS.PPP [24].

## Specification and estimation

We utilized data on reported quantities ($q_{ht}$) and expenditures ($c_{ht}$) for cigarettes consumption as recorded in the household survey across three waves. These data reflect the assumption that households within a specific locality are subject to common market prices during each wave. However, because direct information on cigarette prices is unavailable in the dataset, the Deaton's methodology exploits the commonality of prices within localities, which is implicitly reflected in the $q_{ht}$ and unit values ($v_{ht} = c_{ht}/q_{ht}$).

In Deaton's model, the primary identifying assumption is that inter-cluster price variation serves as an instrumental variable (IV) for demand. This assumption remains valid for our panel data context: while households within a cluster face a uniform price at any given time point, these prices may exhibit temporal variation across waves. Such variation requires careful adjustments in unit values and budget share regressions (Equation 1 and 2) but does not invalidate the underlying identification and estimation strategy discussed below. Based on these information sets and assumptions, we estimated unit values ($v_{ht}$) and budget-share ($w_{ht}$) regression using the following specifications:

$$ln\, v_{hct} \;=\; \alpha^1 + \beta^1 ln\, x_{hct} \; + Z_{hct}\gamma^1 + \psi ln\pi_c + y_t + u^1_{hct} \tag{1}$$

$$w_{hct} = \; \alpha^o + \beta^o ln\, x_{hct} \; + Z_{hct}\gamma^o + \theta ln\pi_c + f_c + y_t + u^o_{hct} \tag{2}$$

In these specifications, $ln\, v_{hct}$ denotes the logarithm of unit values derived in the preceding step and $w_{hct}$ represents the budget share for cigarette consumption: $w_{hct} = c_{hct}/x_{hct}$. The variable $ln\, x_{hct}$ denotes the logarithm of total household consumption, and $Z_{hct}$ represents a vector of household-specific socioeconomic characteristics—such as household size, age and gender composition, and the attributes of household head—which serve as covariates that shift both unit values and budget share. Furthermore, $f_c$ and $y_t$ represent cluster-specific and year specific fixed effects, respectively, and $u^1_{hct}$ and $u^o_{hct}$ denote idiosyncratic error terms in the respective equations. In line with Deaton's original framework, these error terms are assumed to incorporate the measurement errors in both unit values and budget shares. Notably, Equation 1 excludes cluster fixed effects because, as Deaton [13] elaborates, conditional on cross-sectional (and temporal, which we add to accommodate panel set-up) variation in prices, unit values are preliminary driven by quality effects and measurement errors. The term $ln\, \pi_c$ represents unobserved prices, which are assumed to remain constant within clusters once year specific fixed effects are controlled for. We estimated these equations omitting $ln\, \pi_c$ and, later, recovered the associated coefficients using the analytical expressions proposed by Deaton [13]. The relationship between the two error terms, $u^1_{hct}$ and $u^o_{hct}$, was used to correct the price elasticity of demand for cigarettes.

In the pooled OLS set-up of Equations 1 and 2, we account for potential biases that may arise from repeated measurements across the three waves. These biases originate from temporal trends in unit values and budget shares, temporal correlation in the error terms, or unobserved shifts in cigarette consumption patterns that might be correlated with the covariates. We address these issues as follows:

1. We incorporate year-specific fixed effects to ensure that inter-cluster variation in the unit value and budget share regressions is free from temporal trends. This step preserves the fundamental requirements for applying Deaton's demand model.

2. In equation 1 and 2, within-cluster variation may be attributed to: (i) shifts in cigarette consumption preferences over the three-year period, (ii) temporal correlation in the error terms of Equations 1 and 2, and (iii) measurement error (correlation between the error terms of Equations 1 and 2). While (iii) is central to Deaton's correction strategy for the *measurement error*, we minimize bias that may result from (i) and (ii) through the following adjustments:

- We account for temporal correlation in $u_{hct}^1$ and $u_{hct}^o$ by allowing the errors to cluster over time during the estimation of the *measurement error* in the Equation 5.

- We assume that policy-driven shifts in cigarette preferences were minimal during the survey period. The stable, marginal fluctuations in cigarette prices and taxes between 2016 and 2018 (Table 1) support this assumption.

- Cigarette consumption typically exhibits a stable pattern over short intervals. A one-year gap between survey waves makes preference shifts unlikely; evidence from longitudinal studies suggest that transitions between smoking states are rare within such a timeframe [24].

By implementing these assumptions and adjustments, the coefficient estimates from the pooled OLS remain unbiased, and the measurement error corrections retain the structural properties of the original Deaton (1988) model.

Next, using Equations 1 and 2, we derived the demand for cigarettes ($w_{hct}$) and their corresponding unit values ($v_{hct}$) at the cluster level after removing the effects of total household consumption, socio-demographic variables, household head's attributes, and year-specific fixed effects. These residual values were then aggregated and averaged to the cluster level. The primary objective was to estimate elasticity based on cluster-level unit values ($\hat{y}_c^1$) and budget share ($\hat{y}_c^o$). These values capture the commonality of market price and cigarette purchasing patterns within each specific cluster. From equations (3) onwards, the analysis relies on cluster-level averages, having purged the influence of systematic household characteristics and temporal effects. At this stage, the estimation procedure becomes identical to Deaton's standard cross-sectional approach.

$$\widehat{y}_c^1 = \frac{1}{n_c} \Sigma_{h=1}^{n_c} \left( lnv_{hc} - \widehat{\beta}^1 ln\, x_{hc} - Z_{hc}\widehat{\gamma}^1 \right)$$
(3)

$$\widehat{y}_c^0 = \frac{1}{n_c} \Sigma_{h=1}^{n_c} \left( w_{hc} - \widehat{\beta}^o ln\, x_{hc} - Z_{hc}\widehat{\gamma}^o \right)$$
(4)

Using the cluster-level values derived in the previous step, we estimated the regression coefficient of cluster-level demand ($\hat{y}_c^o$) on cluster level unit values ($\hat{y}_c^1$) following the expression in equation 5. We employed a standard errors-in-variables regression to correct for measurement errors by utilizing the covariance and variance of error terms. In this step, we account for correlation among households over time to address the autocorrelation in error terms ($\hat{\sigma}^{10}$) of Equation 1 and 2.

$$\hat{\phi} = \frac{\left( cov\left(\hat{y}_c^1, \hat{y}_c^o\right) - \frac{\hat{\sigma}^{10}}{n_c} \right)}{\left( var\left(\hat{y}_c^1\right) - \frac{\hat{\sigma}^{11}}{n_c^+} \right)}$$
(5)

In the final step, we used a quality correction formula developed by Deaton [13] to derive the price elasticity of cigarette demand ($\hat{\varepsilon}_p$). This step allowed us to isolate genuine price responses from variations in unit values that reflect differences in brand or product quality.

$$\hat{\varepsilon}_p = \left( \frac{\hat{\theta}}{\overline{w}} \right) - \hat{\psi}$$
(6)

In equation 6, $\overline{w}$ represents the sample mean of the share of total household consumption expenditure allocated to cigarettes. The estimated parameters $\hat{\psi}$ and $\hat{\theta}$ correspond to the coefficients on the unobserved cluster-level cigarette prices in Equations 1 and 2, respectively. We recovered these values using the following expressions proposed by Deaton (1988).

$$\hat{\psi} = 1 - \frac{\hat{\beta}^1\left(\overline{w} - \hat{\theta}\right)}{\hat{\beta}^1 + \overline{w}} \tag{7}$$

$$\hat{\theta} = \frac{\hat{\phi}}{1 + \left(\overline{w} - \hat{\phi}\right)\hat{\zeta}} \tag{8}$$

$$\hat{\zeta} = \frac{\hat{\beta}^1}{\hat{\beta}^o + \overline{w}\left(1 - \hat{\beta}^1\right)} \tag{9}$$

We also recovered the expenditure elasticity of demand using the following expression proposed by Deaton (1988),

$$\hat{\varepsilon}_I = 1 + \left(\frac{\hat{\beta}^o}{\overline{w}}\right) - \hat{\beta}^1 \tag{10}$$

We used R-programming for statistical computing for data management and analysis. Specifically, we employed the *tidyverse* package for data management, *fixest* for estimating fixed-effect regression models, and the *survey* package to generate survey-weighted estimates. Additionally, we used the *boot* package to calculate bootstrapped standard errors [25–28].

### Survey data, cluster definitions, and measurement

We utilized data from the Nepal Household Risk and Vulnerability Survey, a three-year (2016–2018) household panel. The survey was conducted based on the protocol approved by Central Bureau of Statistics (CBS) as per the Statistical Act (1958) [29]. Verbal consent was obtained from each respondent after a thorough introduction of the survey. All respondents were briefed about the voluntary nature of participation. Participants were assured that the information they share during the interview will be kept confidential and anonymous. These data were accessed in July 2022 from The World Bank data repository. Detailed survey methodology is provided by Walker et al. (2019) [30]. The survey instrument, which is based on World Bank's Living Standards Measurement Survey, collected information on socio-demographics, consumption expenditure, income, and labor supply. The sampling frame included all households in rural municipalities, municipalities and sub-metropolitan areas of Nepal, stratified across 11 regions. In the first sampling stage, 400 wards were randomly selected, with the number per region proportional to its population share. In the second stage, 15 households were randomly selected per ward, resulting in a total of 6,000 households interviewed in 2016. Attrition remained low: 5,835 households (97.3%) were reinterviewed in 2017, and 5,696 (94.9%) in 2018. A total of 5,654 households (94.2%) completed all three waves. After excluding one household due to missing food expenditure data in 2017, we utilized a balanced, complete-cases sample of 5,653 households, yielding 16,959 observations for analysis.

In this analysis, we utilized clusters defined by the survey design, resulting in a total of 400 clusters. On average, each cluster contains 42.4 observations across the three waves stemming from the follow-up of each household between 2016 and 2018. This approach contrasts significantly with that of Vladisavljevic, Zubović, Đukić and Jovanović in Serbia [31] where clusters were defined as interaction of *clusters* and *waves*. Similarly, Chelwa and van Walbeek [32] conducted separate analyses for each survey wave in Uganda. In contrast, we utilized repeated observations within the clusters for three primary reasons: 1) to increase the cluster sizes, thereby enhancing the robustness of the estimates; 2) to improve

the reliability of unit value computations, as each household reports consumption at three distinct time points; and 3) to increase the estimation efficiency by reducing error variance.

The survey questionnaire is utilized to collect data on primary variables used in this analysis. We utilized weekly household expenditure on cigarettes, the quantity of cigarettes purchased, and total annual household expenditure. To ensure a uniform recall period, annual expenditure was converted into weekly equivalents. All monetary values were adjusted to 2017 prices by using the Consumer Price Index provided by Nepal Rastra Bank [33].

As specified in the Equation 1 and 2, we incorporated several household-level socio-demographic variables, including household size, age and gender composition, number of male members, education level and gender of household head. We also accounted for maximum education level within the household, consumption quintiles, and place of residence. These variables were extracted from Sections 1 and 2 of the survey questionnaires, with the exception of consumption quintiles which were calculated based on total annual per capita consumption expenditure. Because the place of residence of the household was not explicitly categorized in the initial questionnaire, we defined rural households as those located in rural municipalities and urban households as those in municipalities and sub-metropolitan cities.

## 3. Results

Table 2 presents the descriptive statistics of the variables used to estimate Equations 1 and 2. The data, pooled over three years, indicate that nearly 17% households reported cigarette consumption. Regarding expenditure, the mean weekly household consumption expenditure was NPR 3,562.7, with 15% of households report cigarettes consumption. On average, households consumed 42.3 cigarettes per week, spending approximately NPR 170. This results in a unit value of NPR 8.2 per cigarette. The unit value contains information on both market prices and consumer quality preferences in the local market. Finally, cigarette expenditure accounted for 5% of total household consumption.

Table 3 presents the descriptive statistics for the control variables used in Equations 1 and 2. As with previous analysis, these statistics are pooled over three survey waves. The average household consist of 5 members (2.4 males). The mean maximum education within households is seven years. Demographic breakdown by age and sex show 0–5 years–olds (4% male; 3.5% female), 18–49 year–old (13% male; 25% female), and those 50+ (11% for both). Finally, 21% of household are headed by female, who have an average of four years of schooling.

Table 4 presents the first-stage regression results for unit value and budget share regressions in Equations 1 and 2. Column (1) reports the estimated coefficients for the unit value regression, which is estimated conditional on households reporting cigarette consumption. The coefficient for total expenditure is positive and statistically significant, yielding a

**Table 2. Descriptive statistics of the outcome variables.**

| Variables | Observations | Mean | SD | Q10 | Q90 |
|---|---|---|---|---|---|
| Total consumption per week (NPR) | 16959 | 3562.7 | 2514.4 | 1554.9 | 6056.4 |
| Positive expenses on cigarette (= 1) | 16959 | 0.1530 | 0.3600 | 0.00 | 1 |
| Cigarette quantities per week (pieces) | 2756 | 42.3 | 42.0 | 7.0 | 120.0 |
| Expenditure on cigarettes per week (NPR) | 2756 | 169.9 | 342.2 | 40.00 | 350.0 |
| Unit value of cigarette (NPR) | 2756 | 8.2 | 15.9 | 1.8 | 10.0 |
| Budget share of cigarette | 2756 | 0.0514 | 0.0951 | 0.0118 | 0.1030 |
| Number of observations in each cluster ($ht$) | | 42.4 | 3.1 | 39 | 45 |
| Number of households in each cluster ($h$) | | 14.1 | 1.0 | 13 | 15 |
| Number of clusters ($c$) | 400 | | | | |
| Number of clusters ($c$) × years ($t$) | 1200 | | | | |

**Note**: Data pooled over three survey years; survey weights applied. SD: Standard Deviation. Q10/Q90: 10th and 90th percentiles. Prices are in 2017 NPR (NPR1000 = US$9.64).

**Table 3. Descriptive statistics of the control variables.**

| Variables | Observations | Mean | SD | Minimum | Maximum |
|---|---|---|---|---|---|
| *Household demographics* | | | | | |
| Household size | 16959 | 4.9711 | 2.0328 | 1 | 22 |
| Number of male members | 16959 | 2.3617 | 1.2760 | 0 | 10 |
| Maximum education of HH members (years) | 16959 | 6.9137 | 4.5758 | 0 | 15 |
| *Proportion of household members (age group in years)* | | | | | |
| Male [0–5] | 16959 | 0.0406 | 0.0894 | 0 | 0.6667 |
| Female [0–5] | 16959 | 0.0364 | 0.0869 | 0 | 0.6667 |
| Male [6–17] | 16959 | 0.1313 | 0.1627 | 0 | 0.8000 |
| Female [6–17] | 16959 | 0.1212 | 0.1585 | 0 | 1 |
| Male [18–49] | 16959 | 0.1840 | 0.1627 | 0 | 1 |
| Female [18–49] | 16959 | 0.2512 | 0.1550 | 0 | 1 |
| Male [>49] | 16959 | 0.1185 | 0.1549 | 0 | 1 |
| Female [>49] | 16959 | 0.1169 | 0.1659 | 0 | 1 |
| *Household head's characteristics* | | | | | |
| HH head is female (= 1) | 16959 | 0.2115 | 0.4084 | 0 | 1 |
| HH head's education (years) | 16959 | 4.0358 | 4.3579 | 0 | 15 |

**Note**: The statistics are pooled over three survey years. Survey weights are used. SD = Standard Deviation. HH = household

quality elasticity of 0.78. Conversely, the coefficient for household size is negative, indicating a lower unit value with the increase in household size. While socio-demographic composition and household head's characteristics are not statistically significant, their signs align with the theoretical expectations; for example, a higher proportion of male members is associated with lower unit values. The negative and significant coefficients for quintiles indicate that, relative to the lowest quintile, households belonging to all other quintiles use cigarettes with lower unit values. Furthermore, with year dummies, we do not find any statistically significant changes in unit values, even though the coefficients are relatively large and positive. Finally, cluster fixed effects are jointly significant, confirming substantial spatial variation in unit values.

In column (2), we present coefficients for the budget share regression. This is an unconditional model, as it was estimated on the entire sample regardless of whether households reported cigarette consumption. The coefficient for the log of total consumption is positive and statistically significant, indicating that households with higher total expenditure allocate a larger share of their budget for cigarettes. Specifically, a 1% increase in total consumption increases the budget share by 0.6 percentage points. The coefficients for expenditure quantiles are statistically significant and negative. This means that relative to the lowest quintile, wealthier households spend smaller percentage of their budgets on cigarettes. We also find that the budget share decreased over as the coefficients for 2017 and 2018 are negative and statistically significant compared to 2016 levels. Finally, the joint statistical significance of the cluster fixed effects confirms the presence of spatial variation in budget shares.

Table 5 provides the estimate of expenditure and price elasticity derived from the procedure described in Equations 3–10. We use the coefficients, along with the key parameters from equations 1 and 2, to find the expenditure elasticity ($\hat{\varepsilon}_I$). The estimated value is 1.06—a positive, statistically significant value indicating that among the cigarette-consuming households, a 10% increase in total household expenditure is associated with a proportional increase in the quantity of cigarettes smoked.

After estimating the expenditure elasticity and combining it with the estimates from cluster-level averages of budget share and unit values, we derived the price elasticities as specified in Equations 3–6. The results are in row (2) of Table 5. The

**Table 4. Unit-value and budget share regressions for cigarette consumption.**

| Variables | (1) | (2) |
|---|---|---|
| | **Unit values of cigarettes in logs** | **Budget share for cigarettes** |
| Total consumption in logs | 0.7785** (0.1709) | 0.0066* (0.0021) |
| Household size in logs | −0.5774** (0.1318) | −0.0098 (0.0053) |
| (Household size in logs)$^2$ | −0.0059 (0.0739) | 0.0018 (0.0022) |
| **Household's age-sex composition (age-group in years) – baseline: females >49** | | |
| Male [0–5] | −0.2599 (0.2319) | −0.0133* (0.0045) |
| Female [0–5] | 0.1740 (0.2946) | −0.0043 (0.0089) |
| Male [6–17] | −0.1524 (0.1584) | −0.0047 (0.0070) |
| Female [6–17] | −0.0632 (0.1095) | −0.0066 (0.0036) |
| Male [18–49] | −0.1428 (0.0978) | −0.0015 (0.0067) |
| Female [18–49] | 0.0757 (0.1063) | −0.0051** (0.0010) |
| Male [>49] | −0.0529 (0.3726) | 0.0020 (0.0055) |
| **Household head's characteristics** | | |
| HH head (female = 1) | 0.0624 (0.1218) | −0.0035 (0.0017) |
| HH head's education (years) in logs | 0.0229 (0.1260) | 0.0005 (0.0003) |
| (HH head's education (years) in logs)$^2$ | 0.0169 (0.0601) | −0.0004* (0.0001) |
| **Other household characteristics** | | |
| Number of male members | 0.0317 (0.0146) | −0.0007 (0.0013) |
| Maximum education of HH members (years) in logs | −0.0307 (0.1430) | 0.0016 (0.0013) |
| (Maximum education of HH members (years) in logs)$^2$ | −0.0018 (0.0553) | −0.0012 (0.0005) |
| **Consumption quintiles (First – baseline)** | | |
| Second (=1) | −0.2475** (0.0424) | −0.0025* (0.0008) |
| Third (=1) | −0.4552* (0.1165) | −0.0048* (0.0015) |
| Fourth (=1) | −0.5647* (0.1313) | −0.0060 (0.0020) |
| Fifth (=1) | −0.9139** (0.2115) | −0.0089* (0.0026) |
| **Place of residence (Rural – baseline)** | | |
| Urban (=1) | −0.0931 (0.1093) | −0.0018 (0.0006) |
| **Survey year (2016 – baseline)** | | |
| Year – 2017 (=1) | 0.1021 (0.0367) | −0.0030*** (0.0001) |
| Year – 2018 (=1) | 0.0800 (0.0871) | −0.0039*** (0.0003) |
| **Cluster dummies** | **F (377, 2378)** | **F (911, 1844)** |
| | **2.9958*** | **1.7631*** |
| **R$^2$ Adjusted** | **0.262** | **0.037** |
| **Observations** | **2756** | **16959** |

**Note:** OLS estimates of coefficients from models 1 and 2. Standard errors (SEs) in parentheses. The autocorrelation of errors over the years is adjusted. The summary of controls is covered in Table 2. The analysis is based on a panel of 5,653 households observed over three years (2016–2018), giving 16,959 observations. Sample and specifications as in Equations 1 and 2. NPR = Nepalese Rupee, 2017 prices, NPR1000 = US$9.64. HH = household. *, **, *** indicate statistical significance at 10%, 5%, and 1%, respectively.

estimated price elasticity of cigarettes ($\widehat{\varepsilon}_p$) is −0.58, indicating that a 10% increase in cigarette prices would lead to a 5.8% decrease in cigarette demand. A 95% confidence interval was computed using 1,000 bootstrap replications, with the lower and upper bounds of −0.79 and −0.37, respectively.

**Table 5. Elasticity estimates of cigarette demand.**

| | Term | Estimate | SEs | P-values | Confidence interval (95%) | |
|---|---|---|---|---|---|---|
| | | | | | Lower | Higher |
| (1) | Expenditure elasticity ($\hat{\varepsilon}_I$) | 1.0578 | (0.2619) | <0.001 | 0.5445 | 1.5712 |
| (2) | Price elasticity* ($\hat{\varepsilon}_p$) | −0.5759 | (0.1073) | <0.001 | −0.7949 | −0.3745 |

**Note:** Row (1) provides expenditure elasticity using equation 10. Row (2) provides the price elasticity of cigarette demand using equations 6–9. Standard errors (SEs) in parentheses. SEs for price elasticity are based on 1,000 bootstrap replications from equations 6–9 and SEs for expenditure elasticity are based on 1,000 bootstrap replications of the equations 1, 2 and 10.

\* We also undertake robustness checks of $\hat{\varepsilon}_p$ across the three survey panels – the estimates range from −0.15 in 2016 to −0.95 in 2018.

## 4. Discussion and conclusion

This paper aims to estimate the price elasticity of demand for cigarettes using a three-year panel household survey microdata from Nepal. Our results indicate that the demand is inelastic, with price elasticity ranging from −0.79 to −0.37, and an average value of −0.58.

Findings on prevalence and expenditure pattern from Nepal show consistency with those of other South Asian countries and other LMICs [18,19,31,32,34–37]. In Nepal, 15 percent of households report non-zero cigarette expenditure, which is higher than the regional average of 10 percent but falls within the range observed across South Asia, from 4.5 percent in Bhutan to 18.7 percent in the Maldives [19]. Prevalence estimates reported in other settings show substantial variation, ranging from 1.5 percent in Nigeria to seven percent in Uganda, and as high as 40 percent in Bosnia and Herzegovina and Serbia [31,32,34,36,38]. In contrast, expenditure shares show relatively limited variation, with Nepal at 5.14 percent, compared to a range of 3.9 percent in India to 7.6 percent in Uganda [32,37] Similarly, our regression estimates for unit value (0.78) and budget share (0.0067) also fall within the wide range reported in the existing literature across South Asian countries and other LMICs [6,14,17,35,39]. With respect to price elasticity of demand, global evidence suggests considerable heterogeneity, with both elastic and inelastic responses observed. For instance, cigarette demand is found to be inelastic in South Africa (−0.86), Vietnam (−0.165), Croatia (−0.63) and Zambia (−0.2), while elastic demand is reported in Pakistan (−1.06), rural Bangladesh (−1.38), and Bosnia and Herzegovina (−1.366). Although most studies report inelastic demand for cigarettes [17,18,36,39–43], there remains substantial cross-country variation in smokers' responsiveness to cigarette price changes. In high income European countries, the price elasticity estimate of cigarette ranges from −0.67 to −0.24 [44].

### Policy relevance

The price and expenditure elasticity estimates are relevant to the policy makers as they provide idea about the impact of tax on cigarette demand and tax revenues in short and long run, respectively. First, the estimated elasticity, along with its lower and upper bounds, fall within the range of −1–0, indicating that cigarette consumption in Nepal is relatively inelastic. This pattern is consistent with evidence from other countries in the South Asian region. Specifically, a one percent increase in cigarette prices is associated with an approximate 0.6 percent reduction in cigarette demand. Consequently, a tax-increase that raises cigarette prices is likely to reduce cigarette demand only modestly. However, a positive and unitary elastic expenditure elasticity suggests that rising incomes or increases in household expenditure can partially offset the price-induced reduction in cigarette demand, thereby moderating the overall decline in cigarette consumption and reinforcing the persistence of cigarette demand. This suggests that cigarette taxation should be complemented by additional tobacco control measures—such as restrictions on smoking in public places, support for nicotine replacement

therapies etc.—to effectively curb cigarette demand. Similar policy implications apply in other settings with inelastic cigarette demand, including India and Bangladesh.

Second, policymakers are also concerned with the revenue implications of cigarette taxation, as government revenue is directly influenced by consumers' responsiveness to price changes and changes in purchasing power. Given the relatively inelastic demand for cigarettes, increases in excise tax are likely to generate higher tax revenues. When demand is inelastic, consumers are less responsive to price increases, allowing tax revenue to rise despite higher cigarette prices. Here, the positive and unitary expenditure elasticity plays a complementary role by indicating how consumption responds to rising incomes. A positive expenditure elasticity suggests that economic growth or increases in household expenditure may offset some of the consumption-reducing effects of higher prices, thereby sustaining or even expanding the tax base. In Nepal, a recent policy discussion paper on excise taxation in the 2024/25 budget examined the revenue effects of higher cigarette taxes [21]. The paper highlights that experience over the past three years demonstrates that increasing tobacco taxes has been an effective means of raising government revenues.

## Limitations

Our approach to estimating the price elasticity of demand for cigarettes is not without limitations. First, due to the low frequency of reported cigarette consumption, we rely on repeated observations pooled from three-year panel surveys to estimate price elasticity. To address potential autocorrelation, we made adjustments as discussed in methodology section, which helps mitigate the possible bias. Second, unlike many other studies, we were unable to report disaggregated estimates of the price elasticity by policy-relevant variables because of the relatively low incidence of cigarette consumption within each cluster. Despite these limitations, our analysis provides a robust price elasticity estimate that is consistent with the existing literature in the South Asian context and relevant to current policy discussions in Nepal.

**Future research directions.** Future research on cigarette demand would benefit from improved data availability and heterogeneity analysis across policy relevant variables—for example, income levels, rural-urban settings, age groups, and gender. Such analyses would enable the design of more targeted and equitable tax and tobacco control policies. In addition, collecting data at more frequent intervals would allow for the examination of smokers' behavioural response to price changes supporting timely and evidence-based policy decisions. Finally, recent surveys in the South Asian region contains a broader set of questions related to tobacco use and smoking behaviour. Utilizing these datasets could facilitate meaningful inter- and intra-regional comparisons and improve understanding of the shifts in smoking behaviour across different contexts.

## Acknowledgments

We sincerely thank the World Bank and all the individuals involved in collecting and making the panel data available, which has been invaluable for our research.

## Author contributions

**Conceptualization:** Vishnu Prasad Sapkota, Umesh Prasad Bhusal, Nirmal Kumar Raut.

**Data curation:** Vishnu Prasad Sapkota, Umesh Prasad Bhusal.

**Formal analysis:** Vishnu Prasad Sapkota.

**Methodology:** Vishnu Prasad Sapkota, Umesh Prasad Bhusal, Nirmal Kumar Raut.

**Project administration:** Vishnu Prasad Sapkota.

**Software:** Vishnu Prasad Sapkota.

**Writing – original draft:** Vishnu Prasad Sapkota, Umesh Prasad Bhusal, Nirmal Kumar Raut.

**Writing – review & editing:** Vishnu Prasad Sapkota, Umesh Prasad Bhusal, Nirmal Kumar Raut.

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
