## [Decision Letter · Decision Letter 0]

19 Aug 2025

Dear Dr. Sapkota,

Thank you for submitting your manuscript to PLOS ONE. After careful consideration, we feel that it has merit but does not fully meet PLOS ONE’s publication criteria as it currently stands. Therefore, we invite you to submit a revised version of the manuscript that addresses the points raised during the review process.

We look forward to receiving your revised manuscript.

Kind regards,

Gulcan Onel, Ph.D.

Academic Editor

PLOS ONE

Journal Requirements:

https://journals.plos.org/plosone/s/file?id=wjVg/PLOSOne_formatting_sample_main_body.pdf   and  and

Reviewers' comments:

Reviewer's Responses to Questions

**Comments to the Author**

1. Is the manuscript technically sound, and do the data support the conclusions?

Reviewer #1: Yes

Reviewer #2: Partly

2. Has the statistical analysis been performed appropriately and rigorously?

Reviewer #1: Yes

Reviewer #2: No

3. Have the authors made all data underlying the findings in their manuscript fully available?

Reviewer #1: Yes

Reviewer #2: No

4. Is the manuscript presented in an intelligible fashion and written in standard English?

Reviewer #1: Yes

Reviewer #2: Yes

Reviewer #1: Comments

The manuscript presents a novel and policy-relevant study estimating the price elasticity of cigarette demand in Nepal using Deaton's demand model, applied to a three-year panel dataset (2016–2018) from the Nepal Household Risk and Vulnerability Survey (NHRVS). This is the first study to estimate cigarette price elasticity in Nepal, filling a critical gap in the literature for South Asia, a region with high smoking prevalence. The manuscript provides a robust contribution to the field of health economics and tobacco control. However, to strengthen the manuscript, the following suggestions should be considered:

1. The introduction is lengthy and could be more concise, particularly in discussing regional comparisons. Streamline the introduction by condensing regional comparisons into a single paragraph.

2. There is need to provide an overview of the trend of tobacco prices and/or excise tax in Nepal. This should be part of the Introduction section.

3. The numerator for Equation 7 should have a w bar, not an underlined w.

4. The discussion is lengthy and repetitive, particularly in comparing expenditure shares and coefficients across countries. Condense the comparison of expenditure shares and coefficients into a table for clarity and brevity.

5. No mention of future research directions beyond the current limitations.

6. The referencing should be reviewed. For instance, Walker et al. (2019)(27) should be Walker et al. (27). Also, note that there has to be space before an open bracket – review the whole manuscript.

Reviewer #2: The paper is written in standard language and provide interesting insights on to the topic of cigarettes consumption elasticities in Nepal. However significant changes need to be done to address the main weakness of the manuscript - combination of underlying data and methodology used. To be more specific Deaton's model is well suited for the cross-sectional data analysis and independent household observations. There seem to be some efforts from authors in this domain however are not well described and thus not possible to assess in the terms of their appropiratness and violating the main feature of current research in terms of replicability. There is relatively plentifull usage of routine surveys to estimate cigarettes elasticities, but if the authors reliably amend their approach and manuscript accordingly their main weakness can turn into a main contribution.

For robustness of the results it would be prefferable to use also all three waves of surveys to estimate elasticities separately and present results for those as further validation of the approach. Finally, please do a through revision of the literature review as some of the information seem not to fit the contents of the papers cited. For instance reference (14) to John (2008) indicate that for additional products to bidis and leaf tobacco elasticities are estimated, however the paper only investigates cigarettes, bidis and leaf tobacco.

**Do you want your identity to be public for this peer review?** For information about this choice, including consent withdrawal, please see our For information about this choice, including consent withdrawal, please see our Privacy Policy .

Reviewer #1: **Yes:** Chengetai DareChengetai Dare

Reviewer #2: No

While revising your submission, please upload your figure files to the Preflight Analysis and Conversion Engine (PACE) digital diagnostic tool, https://pacev2.apexcovantage.com/ . PACE helps ensure that figures meet PLOS requirements. To use PACE, you must first register as a user. Registration is free. Then, login and navigate to the UPLOAD tab, where you will find detailed instructions on how to use the tool. If you encounter any issues or have any questions when using PACE, please email PLOS at . PACE helps ensure that figures meet PLOS requirements. To use PACE, you must first register as a user. Registration is free. Then, login and navigate to the UPLOAD tab, where you will find detailed instructions on how to use the tool. If you encounter any issues or have any questions when using PACE, please email PLOS at figures@plos.org . Please note that Supporting Information files do not need this step.. Please note that Supporting Information files do not need this step.

---

## [Author Response · Author response to Decision Letter 1]

10 Oct 2025

10/10/2025

To

Gulcan Onel, Ph.D.

Academic Editor

PLOS ONE

Dear Editor,

RE: PONE-D-25-25792 – Price elasticity of demand for cigarettes in Nepal: evidence from a lower middle-income country in South Asia using Deaton’s demand model

Many thanks for considering our paper and for giving us the opportunity to resubmit after addressing the reviewers’ valuable comments. We thank the reviewers for reading the paper carefully and providing comments that have been tremendously helpful in improving it. We include a document that gives a point-by-point response to the comments. We believe that we have addressed all of them.

Please do ask if we have missed anything. We look forward to receiving your decision.

Yours sincerely,

Vishnu Prasad Sapkota

Lecturer

Department of Economics, Tribhuvan University, Nepal Commerce Campus

New Baneshwor, Kathmandu, Nepal

vishnu.sapkota@ncc.tu.edu.np

We thank the reviewers for reading our paper carefully and providing valuable comments that have been tremendously helpful in improving it. This document explains our response to each comment.

Reviewer #1:

Comment 1: The introduction is lengthy and could be more concise, particularly in discussing regional comparisons. Streamline the introduction by condensing regional comparisons into a single paragraph.

Response: Thank you very much for this feedback. We have reviewed the Introduction section carefully and made it more concise. We have condensed the regional comparisons. The changes can be found in the tracked-changes version of the revised manuscript.

Comment 2: There is need to provide an overview of the trend of tobacco prices and/or excise tax in Nepal. This should be part of the Introduction section.

Response: Thank you again for this important suggestion. We have added the following detail in the fourth paragraph of the introduction section.

Table 1 below shows the trend of cigarette prices and taxes in Nepal over the period of 10 years. The cigarette prices have been increasing at constant rate of ~1.00 USD (in PPP) between 2012 to 2022. This means that the household face similar increment in prices before and during the survey period. During the same period, the excise taxes have remained stable between 26 – 30 during the survey period.

Table 1: Cigarette price and tax in Nepal between 2012 and 2022.

2012 2014 2016 2018 2020 2022

Retain price of cigarette per 20 sticks in PPP dollars 3.65 4.69 5.63 6.78 8.38 10.43

Taxes as percentage of cigarette price (%) 29.86 27.79 26 30 27 31.39

Source: https://www.who.int/data/gho/data/themes/topics/raise-taxes-on-tobacco

Comment 3: The numerator for Equation 7 should have a w bar, not an underlined w.

Response: Thank you. We have corrected this.

Comment 4: The discussion is lengthy and repetitive, particularly in comparing expenditure shares and coefficients across countries. Condense the comparison of expenditure shares and coefficients into a table for clarity and brevity.

Response: Thank you for providing opportunity to revise the discussion. We have removed the repetitive points in the discussion and attempted to make the comparison condense. We have revised the section as follows.

The findings related to prevalence and expenditure patterns using data from Nepal exhibit consistency when compared with its South Asian Counterparts as well as other LMICs. While Nepalese households that incur non-zero spending are 15 percent, which is higher than the regional average of 10 percent, the statistics lie within the range from 4.5 percent in Bhutan to 18.7 percent in the Maldives (19). The prevalence reported elsewhere exhibits wide variation, ranging from 1.5 percent in Nigeria to 7 percent in Uganda and 40 percent in Bosnia and Herzegovina and Serbia (28, 29, 31-33). Expenditure shares (5.14 percent for Nepal), on the other hand, exhibit limited variation, from 3.9 percent in India to 7.6 percent in Uganda (18, 29, 33, 34). Likewise, the regression estimates for unit value (0.78) and budget share (0.0066) also fall within the wide range reported in the literature, which varies across the member nations in the region and other LMICs (14, 17, 28, 29, 31, 33, 35, 36). So far as price elasticity of demand is concerned, global evidence suggests wide variation, exhibiting both elastic and inelastic demands. For example, Vietnam (-0.165) and Zambia (-0.2) exhibit inelastic demand, while Pakistan (-1.06), rural Bangladesh (-1.38), and Bosnia and Herzegovina (-1.366) exhibit elastic demand. Most studies generally find inelastic demand for cigarettes (17, 33, 37, 39, 40, 41), but there is substantial cross-country variation in how smokers respond to changes in the price of cigarettes.

Comment 5: No mention of future research directions beyond the current limitations.

Response: As suggested, we have identified areas of future research (See Discussion, Paragraphs 5) by adding following paragraph.

Depending upon the data availability, it may be possible to extend this study, where researchers may consider heterogeneity of the sample population for analysis (for example, based on income levels, rural-urban settings, age groups, and gender). This has the advantage of making tax and tobacco control policies more targeted and fairer. It is also plausible to collect data at frequent intervals and examine the smokers’ behaviour, since this will help make informed and timely policy and behaviour-related decisions. Likewise, new surveys available in the region comprise a broader set of questions related to tobacco and cigarette smoking. It may be meaningful to utilize the datasets from these surveys to examine and make inter- and intra-regional comparisons of the shifts in the behavioural patterns of the smokers.

Comment 6: The referencing should be reviewed. For instance, Walker et al. (2019)(27) should be Walker et al. (27). Also, note that there has to be space before an open bracket – review the whole manuscript.

Response: Thank you. We have now corrected the references.

Reviewer #2:

Comment 1: To be more specific Deaton's model is well suited for the cross-sectional data analysis and independent household observations. There seem to be some efforts from authors in this domain however are not well described and thus not possible to assess in the terms of their appropriateness and violating the main feature of current research in terms of replicability.

Response: Thank you very much for providing us with an opportunity to elaborate on how the extension of Deaton’s approach to panel data does provide useful results after reasonable assumptions and suitable adjustments. In the methodology section (specification and estimation, paragraph 3), we have added the following paragraphs.

In the pooled OLS set-up of equations 1 and 2, we adjust for possible bias that may arise from repeated measurements over three waves, possibly due to temporal trends in unit values and budget shares, temporal correlation in the error terms of the respective equations, and unobserved changes in cigarette consumption patterns that may be correlated with the covariates. We correct for these as follows:

We add year-specific fixed effects to ensure that between-cluster variation in the unit value and budget share regressions is free from temporal trends. This ensures that the basic requirement to apply the Deaton demand model is maintained.

In these equations, within-cluster variation may be due to (i) variation in cigarette consumption preferences over the three-year period, (ii) autocorrelation in the error terms in equations 1 and 2, and (iii) measurement error (correlation between the error terms of equations 1 and 2). Of these, (iii) is key to Deaton’s correction strategy for measurement error. We minimize bias that may result from (i) and (ii) with the following adjustments and assumptions:

We adjust for temporal correlation in u_hct^1and u_hct^oby allowing the errors to cluster over time while estimating the measurement error.

We assume that policy-driven shifts in cigarette preferences are minimal during the survey period – a stable, marginal change in cigarette prices and taxes between 2016–2018 (shown in Table 1) provides evidence that this assumption is met.

Cigarette consumption behaviour exhibits a stable pattern over short time interval. A one-year gap between each survey wave makes preference changes unlikely – longitudinal studies show that transitions between smoker and non-smoker states are rare within one year (24).

Together, with these assumptions and adjustments, the coefficient estimates from the pooled OLS and measurement error corrections still maintain the properties of the original Deaton (1988) model.

Comment 2: For robustness of the results, it would be preferable to use also all three waves of surveys to estimate elasticities separately and present results for those as further validation of the approach.

Response: Yes, we have performed robustness checks in individual years. The estimate ranges -0.15 in 2016 to -0.95 in 2018, and both values indicate inelastic demand. These values are reported in the notes of Table 5 as follows.

We also undertake robustness checks of (ε_p ) across the three survey panels – the estimates range from -0.15 in 2016 to -0.95 in 2018.

Comment 3: please do a thorough revision of the literature review as some of the information seem not to fit the contents of the papers cited. For instance, reference (14) to John (2008) indicate that for additional products to bidis and leaf tobacco elasticities are estimated, however the paper only investigates cigarettes, bidis and leaf tobacco.

Response: Thank you very much for providing an opportunity to revise our literature review and discussion. We have thoroughly revised it, and it is available in the track changes in the text.

---

## [Decision Letter · Decision Letter 1]

19 Jan 2026

Dear Dr. Sapkota,

**As mentioned by the reviewers, the manuscript has greatly improved in depth as well as quality. Though some minor improvements have been suggested including some formatting edits and linguistic corrections.****Kindly review and proofread the document thoroughly and resubmit with mentioned changes.**

plosone@plos.org . . A letter that responds to each point raised by the academic editor and reviewer(s). You should upload this letter as a separate file labeled 'Response to Reviewers'.A marked-up copy of your manuscript that highlights changes made to the original version. You should upload this as a separate file labeled 'Revised Manuscript with Track Changes'.An unmarked version of your revised paper without tracked changes. You should upload this as a separate file labeled 'Manuscript'.

We look forward to receiving your revised manuscript.

Kind regards,

Yogesh Kumar Jain, PhD

Academic Editor

PLOS One

**Journal Requirements:**

Reviewers' comments:

Reviewer's Responses to Questions

**Comments to the Author**

Reviewer #1: All comments have been addressed

Reviewer #2: (No Response)

2. Is the manuscript technically sound, and do the data support the conclusions?

Reviewer #1: Yes

Reviewer #2: Yes

3. Has the statistical analysis been performed appropriately and rigorously?

Reviewer #1: Yes

Reviewer #2: Yes

4. Have the authors made all data underlying the findings in their manuscript fully available?

Reviewer #1: Yes

Reviewer #2: No

5. Is the manuscript presented in an intelligible fashion and written in standard English?

Reviewer #1: No

Reviewer #2: No

**Reviewer #1:** As I indicated previously, there should be space before the reference. For instance, from the first 6 lines of the fist paragraph of the introduction, it is evident that my previous comment was not considered:As I indicated previously, there should be space before the reference. For instance, from the first 6 lines of the fist paragraph of the introduction, it is evident that my previous comment was not considered:

Cigarette smoking is declining globally; however, its prevalence remains highest in South Asia.

Recent evidence report that 27.1% of people aged 15 and above in South Asia smokes

cigarettes, with Nepal reporting an even higher rate of 30.4%, compared to the global average

of 22.3% (1, 2). Cigarette smoking is documented as one of the important modifiable riskfactors

for leading non-communicable diseases(3, 4). Beyond its impact on health, smoking

further aggravates living standards and poverty(5-7).

The authors should review the whole manuscript, accordingly.

**Reviewer #2:** The manuscript significantly improved on depth and quality of explaining the process. The use of proofreading service is strongly suggested. Some minor comments to be addressed are made in the text of the manuscript.The manuscript significantly improved on depth and quality of explaining the process. The use of proofreading service is strongly suggested. Some minor comments to be addressed are made in the text of the manuscript.

**Do you want your identity to be public for this peer review?** For information about this choice, including consent withdrawal, please see our For information about this choice, including consent withdrawal, please see our Privacy Policy .

Reviewer #1: No

Reviewer #2: No

---

## [Author Response · Author response to Decision Letter 2]

15 Feb 2026

Editor’s comments:

[Comment E0] Thank you for including your ethics statement on the online submission form: "The analysis is based on the survey data that was conducted based on the protocol for the survey was approved by the Central Bureau of Statistics (CBS) as per the Statistical Act (1958) in September 2018. Verbal consent was obtained from each respondent after a thorough introduction of the survey. All respondents were briefed about the voluntary nature of participation. Participants were assured that the information they share during the interview will be kept confidential and anonymous.".

To help ensure that the wording of your manuscript is suitable for publication, would you please also add this statement at the beginning of the Methods section of your manuscript file.

[Response] Thank you for pointing out the missed opportunity to clarify the ethical aspects in the paper. Now we have included it in page 11/12, section Methodology/ Survey data, cluster definitions, and measurement, line 2-6.

"The survey was conducted based on the protocol approved by Central Bureau of Statistics (CBS) as per the Statistical Act (1958) (29). Verbal consent was obtained from each respondent after a thorough introduction of the survey. All respondents were briefed about the voluntary nature of participation. Participants were assured that the information they share during the interview will be kept confidential and anonymous."

[Comment E1] If the reviewer comments include a recommendation to cite specific previously published works, please review and evaluate these publications to determine whether they are relevant and should be cited. There is no requirement to cite these works unless the editor has indicated otherwise.

[Response] Thank you. We did not receive any such recommendations.

[Comment E2] Please review your reference list to ensure that it is complete and correct. If you have cited papers that have been retracted, please include the rationale for doing so in the manuscript text, or remove these references and replace them with relevant current references. Any changes to the reference list should be mentioned in the rebuttal letter that accompanies your revised manuscript. If you need to cite a retracted article, indicate the article’s retracted status in the References list and also include a citation and full reference for the retraction notice.

[Response] Thank you. We have carefully updated the reference list. Below, we have listed the references that were added, updated and removed from the reference list, along with the justification.

1. We added the following two citations: the main purpose of adding references is to provide the sources of price and tax data we covered in Table 1.

a. WHO. Tobacco retail price data 2024 [updated 2024–12–16. Available from: https://apps.who.int/gho/data/view.main.TOBRETAILv.

b. United Nations. PPP conversion factor, GDP (LCU per international $) 2025 [updated 21/02/2025. Available from: https://data.un.org/Data.aspx?d=WDI&f=Indicator_Code%3APA.NUS.PPP.

2. Two papers that were cited in the original submission but missed in the second revision are following, and we have cited these papers again in this revision.

a. Dare C, Boachie MK, Tingum EN, Abdullah S, van Walbeek C. Estimating the price elasticity of demand for cigarettes in South Africa using the Deaton approach. BMJ Open. 2021;11(12):e046279.

b. Barać ŽA, Burnać P, Rogošić A, Šodan S, Vuko T. Cigarette price elasticity in Croatia–analysis of household budget surveys. Journal of Applied Economics. 2021;24(1):318–28.

3. The following two papers were found to be duplicated in the reference list. We have removed the duplication.

a. John RM. Price elasticity estimates for tobacco products in India. Health Policy and Planning. 2008;23(3):200–9.

b. Huque R, Abdullah SM, Hossain MN, Nargis N. Price elasticity of cigarette smoking in Bangladesh: evidence from the Global Adult Tobacco Surveys (GATS). Tobacco Control. 2024;33(Suppl 2):s51–s8.

4. We did not remove any paper that were originally cited. The final list contains 43 references, while original submission had 41 references.

Reviewer 1 comments:

[Comment R1.1] As I indicated previously, there should be space before the reference. For instance, from the first 6 lines of the fist paragraph of the introduction, it is evident that my previous comment was not considered:

Cigarette smoking is declining globally; however, its prevalence remains highest in South Asia. Recent evidence report that 27.1% of people aged 15 and above in South Asia smokes cigarettes, with Nepal reporting an even higher rate of 30.4%, compared to the global average of 22.3% (1, 2). Cigarette smoking is documented as one of the important modifiable riskfactors for leading non-communicable diseases(3, 4). Beyond its impact on health, smoking further aggravates living standards and poverty(5-7).

The authors should review the whole manuscript, accordingly.

[Response] Thank you. We have thoroughly revised the manuscript to address the reviewer’s suggestions on citation format.

Reviewer 2 comments:

The manuscript significantly improved on depth and quality of explaining the process. The use of proofreading service is strongly suggested. Some minor comments to be addressed are made in the text of the manuscript.

[Response:] Thank you for providing us an opportunity to clarify many of our statistical and conceptual approaches. We have thoroughly proofread the manuscript and made the suggested corrections. We have also addressed the comments made in the text of the manuscript.

[Comment R2.1] consult the consumption of cigarettes / tobacco in Balkan coutnries and the asia is only fraction of it, need to be amended

[Response] Thank you for your suggestion. We agree that cigarette smoking is very high in Balkan countries. However, we prepared this manuscript in the context of South Asia. Therefore, we did not mention any reference to Balkan countries in the opening sentence. However, we discussed the cigarette smoking behaviour and papers on elasticities from the Balkan countries in the discussion section (page no 18, paragraph 3, line 6 onward).

[Comment R2.2] suggest to reword the sentence to make it clearer if the reference to South Asia is related to Nepalese values or if South Asia is compared to the global values.

[Response] Thank you for pointing out the ambiguity. We have now revised the opening sentence to make it clear to the reader. We have made the following revision in page 4, para 1 line 2-5.

Cigarette smoking is declining globally; however, its prevalence remains highest in South Asia. Recent evidence reports that 27.1% of people aged 15 and above in South Asia smoke cigarettes—exceeding the global average of 22.3%—with Nepal reporting an even higher prevalence of 30.4% (1, 2).

[Comment R2.3] not necesserily given the conficdence interval of expenditure elasticity, suggest tomodificate the wording to eg more responsive to income growth

[Response] Thank you. We have changed the interpretation in the abstract and finding sections (page 2, abstract/results and page 17, last paragraph) of the document. These changes are available in the track changes.

[Comment R2.4] no need to manetion negatgive sign of the coefficient

[Response] Thank you. We have made the suggested corrections.

[Comment R2.5] brackets should be used across the paper for introducing the citations

[Response] Thank you. We have followed the recommended citation format for the Plos One.

[Comment R2.6] add also the line with NPR for reference to the survey based unit values in the analysis

[Response] Thank you. We have added the equivalent NPR values in page 6, Table 1, row 2.

Year 2012 2014 2016 2018 2020 2022

Retail price (per 20 sticks in PPP dollars) 3.65 4.69 5.63 6.78 8.38 10.43

Retail price (per 20 sticks in equivalent NPR*) 95 130 176 211 266 342

Taxes as percentage of cigarette price (%) 29.86 27.79 26 30 27 31.39

Source: The data was accessed from https://www.who.int/data/gho/data/themes/topics/raise-taxes-on-tobacco (23).

*The PPP conversion factors for NPR were obtained from

https://data.un.org/Data.aspx?d=WDI&f=Indicator_Code%3APA.NUS.PPP (24)

[Comment R2.7] it is not clear how this happens in the cluster-level setup where household level charactreristics are averaged out

[Response] the correlation between error terms of Equation 1 and 2, σ ^10, that appears in Equation 5 is derived from the panel data at household level regression. We corrected the autocorrelation issue when deriving the estimate of σ ^10, which is a panel regression. We have improved the sentence for clarity on this issue by modifying the sentence in page 10, line 17-19.

In this step, we account for correlation among households over time to address the autocorrelation in error terms (σ ^10) of Equation 1 and 2.

[Comment R2.8] suggest to elaborate also on the expenditure (often used as income proxy) elasticity as it also play crucial role in forming demand decision and subsequently budget revenues and related health implications

[Response] Thank you for the helpful comments. We have added the following lines in the policy relevance section (page: 19, section: policy relevance, line: 8-12; page: 20, line: 8-12)

However, a positive and unitary elastic expenditure elasticity suggests that rising incomes or increases in household expenditure can partially offset the price-induced reduction in cigarette demand, thereby moderating the overall decline in cigarette consumption and reinforcing the persistence of cigarette demand.

Here, the positive and unitary expenditure elasticity plays a complementary role by indicating how consumption responds to rising incomes. A positive expenditure elasticity suggests that economic growth or increases in household expenditure may offset some of the consumption-reducing effects of higher prices, thereby sustaining or even expanding the tax base.

---

## [Decision Letter · Decision Letter 2]

8 Mar 2026

Price elasticity of demand for cigarettes in Nepal: evidence from a lower middle-income country in South Asia using Deaton’s demand model

PONE-D-25-25792R2

Dear Dr. Sapkota,

We’re pleased to inform you that your manuscript has been judged scientifically suitable for publication and will be formally accepted for publication once it meets all outstanding technical requirements.

Kind regards,

Yogesh Kumar Jain, PhD

Academic Editor

PLOS One

Additional Editor Comments (optional):

Reviewers' comments:

Reviewer's Responses to Questions

**Comments to the Author**

Reviewer #1: All comments have been addressed

Reviewer #2: All comments have been addressed

2. Is the manuscript technically sound, and do the data support the conclusions?

Reviewer #1: Yes

Reviewer #2: Yes

3. Has the statistical analysis been performed appropriately and rigorously?

Reviewer #1: Yes

Reviewer #2: Yes

4. Have the authors made all data underlying the findings in their manuscript fully available?

Reviewer #1: Yes

Reviewer #2: No

5. Is the manuscript presented in an intelligible fashion and written in standard English?

Reviewer #1: Yes

Reviewer #2: Yes

Reviewer #1: (No Response)

Reviewer #2: I have no additional comments for authors at this point. The current version of the manuscript represent significant improvement compared to the original.

**Do you want your identity to be public for this peer review?** For information about this choice, including consent withdrawal, please see our For information about this choice, including consent withdrawal, please see our Privacy Policy .

Reviewer #1: No

Reviewer #2: No

---

## [Editor Report · Acceptance letter]

PONE-D-25-25792R2

PLOS One

Dear Dr. Sapkota,

I'm pleased to inform you that your manuscript has been deemed suitable for publication in PLOS One. Congratulations! Your manuscript is now being handed over to our production team.

Kind regards,

on behalf of

Dr. Yogesh Kumar Jain

Academic Editor

PLOS One